# Diff-TTSG: Denoising probabilistic integrated speech and gesture synthesis

*Shivam Mehta, Siyang Wang, Simon Alexanderson, Jonas Beskow, Éva Székely, Gustav Eje Henter*

Division of Speech, Music and Hearing, KTH Royal Institute of Technology, Sweden

`{smehta, siyangw, simonal, beskow, szekely, ghe}@kth.se`

## Abstract

With read-aloud speech synthesis achieving high naturalness scores, there is a growing research interest in synthesising spontaneous speech. However, human spontaneous face-to-face conversation has both spoken and non-verbal aspects (here, co-speech gestures). Only recently has research begun to explore the benefits of jointly synthesising these two modalities in a single system. The previous state of the art used non-probabilistic methods, which fail to capture the variability of human speech and motion, and risk producing oversmoothing artefacts and sub-optimal synthesis quality. We present the first diffusion-based probabilistic model, called Diff-TTSG, that jointly learns to synthesise speech and gestures together. Our method can be trained on small datasets from scratch. Furthermore, we describe a set of careful uni- and multi-modal subjective tests for evaluating integrated speech and gesture synthesis systems, and use them to validate our proposed approach.

**Index Terms**: Text-to-speech, speech-to-gesture, joint multimodal synthesis, deep generative model, diffusion model, evaluation

## 1. Introduction

Face-to-face (i.e., embodied) human communication involves both spoken and non-verbal aspects. The latter include gaze, facial expression, proxemics, and *co-speech gestures* – head, arm, hand, and body motions that co-occur with speech. The speech and gesture modalities are closely linked and originate from a shared representation of the message the speaker intends to convey [1]. Gestures may complement or supplement the spoken words, or even replace words entirely [2, 1]. Crucially, the presence of gestures have been shown to enhance both human-human [3] and human-machine communication [4, 5]. For this reason, the automatic synthesis of speech and gestures are considered key enabling technologies for Embodied Conversational Agents (ECAs) such as virtual avatars and social robots.

Despite the common origin of human spoken and non-verbal communication, speech synthesis and gesture generation have hitherto largely been treated as separate problems by non-overlapping research communities. Text-to-speech research and data has historically focussed on speech read aloud by a voice actor, whilst human gesticulation is associated with spontaneous speech uttered in conversation. Combining read-style text-to-speech (TTS) with spontaneous speech-to-gesture (STG), using two different systems typically trained on data from different actors results in incoherent expression in the ECA [6]. It may also degrade gesture quality, due to mismatch between the natural speech audio used during STG training and the synthetic audio used to drive gesticulation at synthesis time [7].

Noticing the above discrepancies, there have been a few proposals to integrate the synthesis of both modalities into one single text-driven system [8, 7], a problem termed *integrated speech and gesture synthesis*, or ISG. This is compelling not only because it better resembles the intertwined manner in which communicative behaviour is generated in humans, but also because gesture-motion synthesis and speech-synthesis acoustic modelling are mathematically very similar problems: both may accept text as input and both produce a sequence of continuous-valued vectors as output. They may thus be modelled using similar approaches. ISG also eliminates redundancies, enabling more compact systems that are faster to run [7].

However, unlike the synthesis of read-aloud speech from text – where TTS can reach similar naturalness scores as recorded human speech [9] – spontaneous speech synthesis and the synthesis of gesture motion both present more challenging modelling problems. Approaches that learn to solve both problems simultaneously are still in their infancy, providing an exciting research target. The challenges are both due to the scarcity of high-quality data (e.g., spontaneous speech is rarely captured in controlled conditions; accurate 3D motion data requires marker-based motion capture) and due to the wide variety of different behaviours and expressions in such data; cf. [10]. An accurate description of the full spectrum of human embodied-communication behaviour thus calls for probabilistic approaches based on powerful deep generative models.

In this paper, we propose the first diffusion model for learning to synthesise speech audio and body gestures together from text. Unlike the previous ISG state of the art, our approach is probabilistic, non-autoregressive, and, importantly, can be trained on small datsets from scratch. This removes previous needs for pre-training on large speech datasets and multi-stage training with parts of the network frozen. We perform an in-depth evaluation of our proposal using both uni-modal and multimodal subjective tests, disentangling synthesis quality in different modalities from their appropriateness for each other. Our results find significant improvements over the prior state of the art in all aspects studied. For video examples and code, please see https://shivammehta25.github.io/Diff-TTSG/.

## 2. Related work

We here review related work in speech and gesture synthesis as well as their combination, with a special focus on the use on diffusion models for these tasks.

### 2.1. Speech synthesis

Neural models of acoustics and waveforms have taken TTS technology to new heights [11, 12]. By now, a large number of deep generative approaches have been applied to speech synthesis tasks [13], including GANs [14, 15], VAEs [16], and nor-

malising flows [17, 18]. For large read-speech datasets, TTS naturalness ratings may rival recorded human speech [9].

Most recently, diffusion probabilistic models (DPMs) have risen to prominence in generative modelling following impressive results in text-driven image generation. DPMs models essentially provide a way to model the entire probability distribution of data using only squared error minimisation during training. In speech synthesis, DPMs been used for acoustic modelling, waveform generation (e.g., [19, 20]), and even end-to-end synthesis (e.g., [21]). We here focus on acoustic modelling, as it is the part of TTS that most resembles gesture generation and thus offers a natural integration point. Off-the-shelf neural vocoders can then be used to generate waveforms.

For acoustic models, Diff-TTS [22] was the first work to apply diffusion-based models to synthesise mel-spectrograms. Later, Grad-TTS [23] (described further in Sec. 3.1) formulated the diffusion process as a stochastic differential equation (SDE). Following Grad-TTS, Grad-StyleSpeech [24] introduced reference style vectors for few-shot synthesis, whilst Guided-TTS [25] considered few-shot synthesis with speaker embeddings.

We base our work on Grad-TTS [23] and use the same monotonic alignment search to jointly learn the alignment between input text and synthesised speech and gesture.

## 2.2. Gesture synthesis

Like speech synthesis, data-driven 3D gesture motion generation has made great strides in recent years by leveraging deep-learning-based modelling techniques [5]. Some of these gesture-generation efforts have used deep generative methods, such as VAEs [26, 27] or normalising flows [28]. Systems based on these approaches have exhibited strong performance in recent large-scale gesture-generation challenges [29, 30, 31].

The first examples of gesture generation using diffusion models are very new [32, 33, 34]. Among these, "Listen, Denoise, Action!" [32] adapted the DiffWave [20] architecture and Conformers [35] to the task of audio-driven gesture motion synthesis in different styles (along with demonstrations of dance-motion synthesis and locomotion). GestureDiffuCLIP [34] integrated CLIP [36] into gesture generation to enable the style of the motion to be specified through text, and also to transfer style from arbitrary video clips onto the generated gestures.

## 2.3. Joint verbal and non-verbal synthesis

As described in the introduction, despite the success of deep learning in both speech synthesis and gesture generation, very few works have considered generating both modalities simultaneously. There have been attempts to generate other non-verbal modalities together with speech by using TTS techniques. A prominent example is DurIAN [37], but that system considered speech and facial expression, rather than 3D body motion as here, and did not use spontaneous speech data.

Deep learning for ISG with co-speech body gestures was introduced by Wang et al. [7], which described two different systems for the task. Both were trained on a spontaneous speech dataset [38]. One system was based on a modified version of the Tacotron 2 [12] spectrogram-prediction architecture. Getting that approach to work required first training the speech-synthesis part of the system on a large speech dataset, and then freezing part of the network. Their second approach was a probabilistic ISG obtained by simply training Glow-TTS [17] to concatenated acoustics and pose feature vectors. Unfortunately, whilst that system was able to learn to speak intelligibly, the results were not impressive, and the system was excluded from the majority of their experiments due to poor synthesis quality.

To our knowledge, no prior work exists that uses diffusion models to simultaneously generate speech audio and gesture motion. This is the contribution of this paper, in the process obtaining results that surpass [7], the previous state of the art.

## 3. Method

The task in this paper is to generate a sequence of $T$ acoustic feature vectors, $x_{1:T}$, together with a synchronised sequence of matching 3D poses $g_{1:T'}$ for a speaking and gesturing character, given a sequence of text-derived features $s_{1:P}$, such as phonemes extracted by a TTS front-end, as input. Although acoustics and motion data may have different frame rates in practice (often around 80 fps for acoustics and 30 or 60 fps for the motion), we will assume that the motion has been upsampled/interpolated to match the audio frame rate for the purposes of modelling (so that $T' = T$), with model output resampled to the desired fps for playback.

In this section, we first introduce Grad-TTS [23], and then describe our modifications to Grad-TTS in order to also be able to generate gesture motion along with the speech. The presentation assumes some familiarity with diffusion probabilistic models; for more background on those see [23].

### 3.1. Grad-TTS

Grad-TTS is a text-to-speech system that uses diffusion probabilistic models to sample output acoustics, specifically mel-spectrograms. Accepting a sequence $s_{1:P}$ of $P$ symbols (e.g., phonemes or sub-phonemes) as input, it comprises three parts:

1. an *encoder* that predicts the average mel-spectrum for every input symbol, $\widetilde{\mu}_{1:P} = \mathrm{Enc}(s_{1:P})$;

2. a *duration predictor* that uses $\widetilde{\mu}_{1:P}$ to predict the log-duration $\ln \widehat{d}_{1:P} = \mathrm{DP}(\widetilde{\mu}_{1:P})$ associated with each symbol; and

3. a U-Net [39] *decoder*, $\widehat{x}_{1:T} = \mathrm{Dec}(\widetilde{y}_{1:T}, \mu_{1:T}, n)$, trained to denoise noisy mel-spectrograms $\widetilde{y}_{1:T}$. This is the diffusion probabilistic model.

The encoder predicts a sequence of mean vectors $\widetilde{\mu}_{1:P}$, one for each symbol in the input sequence (usually two symbols per phoneme). By duplicating each vector $\widetilde{\mu}_p$ a variable number of times that correspond to the duration of each (sub-)phone, $\widetilde{\mu}_{1:P}$ is upsampled to obtain a rough approximation $\mu_{1:T}$ of the natural target mel-spectrogram $y_{1:T}$.

To learn to align during training, Grad-TTS uses monotonic alignment search [17], a dynamic-programming algorithm similar to the Viterbi algorithm in HMM training. This procedure identifies the optimal durations (upsampling numbers) $d_{1:P}$ that minimise the squared error between $\mu_{1:T}$ and $y_{1:T}$. The duration predictor is then trained to minimise its mean-squared error in predicting the logarithm of these optimal durations, $\ln d_{1:P}$. During synthesis, $\widetilde{\mu}_{1:P}$ is upsampled based on the output of the duration predictor to obtain $\mu_{1:T}$.

Grad-TTS used the same encoder and duration predictor architecture as Glow-TTS [17]. The main difference from Glow-TTS is the U-Net decoder in Grad-TTS, which defines a diffusion probabilistic model conditioned on $\mu_{1:T}$ that is used to generate samples from the data distribution. This network is trained using a score-matching framework derived from [40]. In essence, noisy mel-spectrograms $\widetilde{y}_{1:T}$ are created by interpolating between the natural target mel-spectrogram $y_{1:T}$ and samples from $\mathcal{N}(\mu_{1:T}, I)$. Dec is trained to minimise the mean squared error in predicting the original $y_{1:T}$ from these noisy

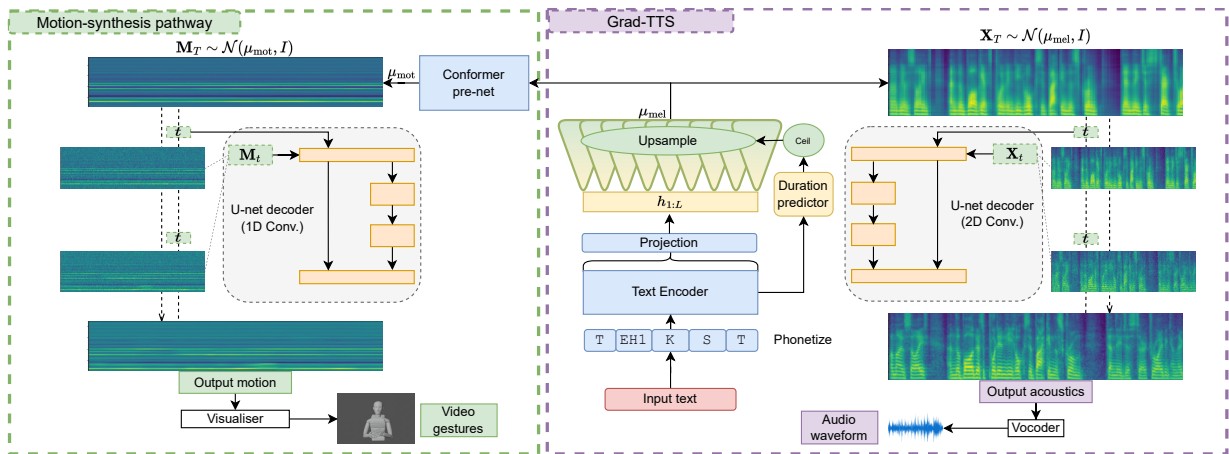

Figure 1: *Overview of the proposed system, illustrating information flow at synthesis time.*

examples. Trained in this way, Dec can be seen as defining an SDE that converts samples from $\mathcal{N}(\mu_{1:T}, I)$ to samples from the natural distribution of mel spectra. For mathematical details, please see [41, 23].

In [23], the U-Net uses the same basic architecture (with 2D CNNs) as the image models in [42], effectively treating mel spectrograms as 2D images. In addition to $\widetilde{y}_{1:T}$, the U-Net is also conditioned on $\mu_{1:T}$ and the amount of noise added (representing the "time dimension" $n$ of the SDE). They also show that using $\mathcal{N}(\mu_{1:T}, I)$ for the noise source enables learning the same distributions as the more conventional choice $\mathcal{N}(0, I)$, but gives better results in practice [23].

Synthesis from the learnt diffusion model amounts to numerically solving the SDE defined by Dec, e.g., using a first-order Euler scheme. This can require many discretisation steps and thus be time consuming. For this reason, Grad-TTS uses an ordinary differential equation (ODE) re-formulation of the SDE denoising process due to [40] when drawing samples. The ODE describes the same target distribution as the learnt SDE, but has better numerical properties and gives superior output quality when solved using a coarse discretisation in order to generate samples more quickly [23].

### 3.2. Modelling speech and motion with Diff-TTSG

Mathematically, motion is represented as a sequence $g_{1:T}$ of *poses* $g_t$ for a 3D character. The numbers in the vector $g_t$ represent quantities such as the translation and rotation of the "root node" of the character (typically the hip bone) in 3D space, along with the rotation of different joints on the character's skeleton. These rotations can be parameterised using, e.g., Euler angles or the exponential map [43]. By bending the joints on a 3D character model according to the rotations specified in $g_t$, a specific pose is obtained. This is similar to how a visual artist may bend joints to pose a wooden articulated mannequin.

Although one may in principle train a TTS model on stacked/concatenated pose-and-acoustics vectors $[g_t^{\mathsf{T}}, x_t^{\mathsf{T}}]^{\mathsf{T}}$, this does not work well in practice, neither using Glow-TTS (as described in [7]), nor with Grad-TTS. For example, the U-Net architecture used by Grad-TTS assumes that feature-vector dimensions are divisible by four, which is rarely the case with pose representations. Simply padding the combined features with additional values until the vector dimensionality was divisible by four led to jittery, low-quality gestures.

To obtain a good model of speech and gestures together, we made a number of changes to the Grad-TTS architecture,

resulting in the architecture illustrated in Fig. 1. Specifically, we incorporated a Conformer "pre-net" [35] to map the upsampled decoder output $\mu_{1:T}$ to corresponding mean predictions $\mu'_{1:T}$ for the pose features. We also added a separate denoising path (left side of the figure) responsible for turning noise sampled from $\mathcal{N}(\mu'_{1:T}, I)$ into convincing pose sequences $g_{1:T}$.

We call the resulting, proposed model *Diff-TTSG*, for *diffusion-based text-to-speech-and-gesture*. The new and old components can be trained jointly in exactly the same way as Grad-TTS, with the loss terms for both the acoustics and gesture diffusion pathways summed.

For our experiments, we adopted the same U-Net architecture in the pose-synthesis path as used for synthesising acoustics (i.e., that of Grad-TTS), except that we replaced all 2D convolutions with 1D convolutions along the time dimension $t$. This is important since, unlike nearby frequency bins on a mel scale, the individual features in the pose representation $g_t$ have no simple spatial relationship, and thus do not possess the approximate translation invariance that make convolutions a good fit for images and mel-spectrograms. The pose-vector visualisations in Fig. 1, thin horizontal streaks, are a reflection of this. Changing to 1D convolutions also means that the dimensionality of $g_t$ no longer has to be evenly divisible by four.

We also experimented with using the alternative U-Net architecture from WaveGrad [19] in the gesture-side U-Net. However, this led to less natural-looking gestures with outstretched arms like a T-pose, which is the origin in joint-rotation space.

## 4. Experiments

We now detail the uni-modal and multimodal perceptual evaluations we performed to validate our proposed method, and the results we obtained. The evaluation design builds on the best practises from speech [44] and motion synthesis [29, 30, 31]. Compared to the recent prior paper on ISG [7], our evaluation more carefully controls for the effect of synthesis quality when assessing how appropriate the different modalities are for each other. Example stimuli are available on the project webpage https://shivammehta25.github.io/Diff-TTSG/.

### 4.1. Data

We evaluated our proposed ISG approach using the Trinity Speech-Gesture Dataset II (TSGD2)[1] [45, 46]. TSGD2 builds

---

[1] https://trinityspeechgesture.scss.tcd.ie/

on the initial Trinity Speech-Gesture Dataset [47] used in previous ISG work [6, 7], but is larger, uses another speaker, and has more accurate mocap.

The TSGD2 dataset comprises six hours of time-aligned 44 kHz audio and 120 fps marker-based motion-capture recordings of a male actor, native in Hiberno English, speaking and gesturing freely and spontaneously to a person situated behind the camera. More detail on the data is given in [45, 46]. We held out 1.5 hours of data for testing and validation, and trained on the remaining 4.5 hours. This is a quite small amount of material compared to the speech corpora often used to train contemporary TTS systems, with the widely used LJ Speech dataset comprising around 24 hours of audio, for example.

Audio was segmented into breath groups like in [48] and then transcribed using Whisper ASR [49]. The same 80-dim. acoustic features as HiFi-GAN were used [15]. The motion data (45-dim. pose vectors $g_t$ with rotations represented using the exponential map [43] as in [28]) was downsampled to 86.13 fps using cubic interpolation to match the frame rate of the acoustic features. Fingers were ignored since finger mocap is notoriously unreliable and prone to unnatural-looking artefacts.

### 4.2. Systems trained

We trained a number of systems on the TSGD2 data, both a unimodal TTS system and several systems capable of generating audio and motion together. Specifically, we trained Grad-TTS on the text-and-audio data using the official source code[2] and default hyperparameters. Training was run for 350k updates with a batch size of 32 and learning rate 1e-4. CMUdict[3] was used to phonemise text. We call this condition **G-TTS**.

We also trained the proposed Diff-TTSG approach on the full multimodal data, based on an extension of the Grad-TTS code and using the same hyperparameters and stopping criterion as for G-TTS. For the Conformer pre-net we used Conformer blocks[4] with 4 hidden layers of 384 channels and 1D filters of length 21 in the convolutional layers, and for the gesture-generation decoder U-Net we used 1D convolutions with 256 hidden channels and kernels of length 5. Synthesis used 50 diffusion steps for speech and 500 for motion. We label output from the trained system **D-TTSG**.

We compared the proposed method both to held-out natural speech and mocap (condition **NAT**) and to a baseline system based on the most successful approach in previous ISG work [7], constituting the previous state of the art on the task. The latter uses a modified version of the Tacotron 2 architecture [12], with an additional gesture-prediction LSTM operating on at the output of the attention-LSTM layer of the decoder. We trained the baseline on the same data as the proposed system (no differences in pre-processing), using the official ISG implementation[5] and the best training protocol identified by [7], which has several stages: starting from a unimodal Tacotron 2 TTS checkpoint[6] trained on the read-aloud LJ Speech dataset[7], the system was fine-tuned on the TSGD2 training audio for 80k updates, before freezing all TTS-related parts of the network and training the remaining gesture-generation weights on the motion from TSGD2 for another 100k updates, using a combined

[2] https://github.com/huawei-noah/Speech-Backbones
[3] http://www.speech.cs.cmu.edu/cgi-bin/cmudict
[4] From https://github.com/lucidrains/conformer
[5] https://github.com/swatsw/isg_official
[6] https://github.com/NVIDIA/tacotron2
[7] https://keithito.com/LJ-Speech-Dataset/

MSE and GAN loss. The GAN loss was based on the output of a discriminator network that takes both speech and gesture as input, with the discriminator predicting whether a given input speech-gesture pair was synthesised or taken from the training data. (Starting from a pre-trained model was necessary because spontaneous speech is difficult for models like Tacotron 2 to learn from scratch, especially from smaller datasets, since these models do not force monotonic alignments between acoustics and phones; cf. [50].) We label the resulting system **T2-ISG**.

We also created an ablation of our model with the same architecture but with modalities trained sequentially instead of jointly. Specifically, we took the trained G-TTS system, froze its weights, added the gesture-synthesis pathway and trained that part for 300k updates. We call this G-TTS+M as the TTS part is the exact same, now just generating motion on top.

### 4.3. Evaluation setup

15 segments were excerpted from the held-out data, selected to be easy-to-comprehend standalone semantically coherent units [51] to be used in the evaluation from episodes 3 and 4 of the corpus. Each segment was approximately 10 seconds long, which has been adequate for accurate evaluation of gesture synthesis in previous studies [28, 7]. Text transcriptions of these phrases were used to synthesise output (acoustics and possibly motion) from the different models in the evaluation.

A pre-trained HiFi-GAN (model `UNIVERSAL_V1`[8]) was used to synthesise waveforms for all stimulus audio, and to create vocoded but otherwise natural speech audio for NAT. We used the avatar from [28, 29] to visualise motion, same as in [7]. This shows a waist-up, skinned 3D character with a fixed hip position and a fixed, lightly cupped pose for each hand. The absence of gaze, lips, facial expression, and lower body are deliberate, since these aspects are not synthesised. Still images of the avatar are seen in Fig. 2, with video examples provided on our project page.

Our subjective evaluations used an interface with five different response options from which only one could be chosen, although the specifics differed slightly in the different studies (see below). In all cases, we recruited native English-speaking participants using the Prolific crowdsourcing platform[9]. For statistical analysis, responses were assigned integer values and tested for significance at the 0.05 level using pairwise $t$-tests.

Similar to [30], four stimuli with embedded attention checks were inserted per participant and study, to filter out unreliable test-takers. The checks instructed participants to give a specific response to the stimulus, either using TTS (for audio-only stimuli), or a text overlay (in video-only stimuli), or an equal mixture of the two (in multimodal stimuli). Each person who completed a study (failing at most one attention check) was remunerated 4.0 GBP for a median of approximately 15 minutes of work, and their responses included in the statistical analysis.

### 4.4. Speech-only evaluation

To evaluate the naturalness of the audio modality of the different conditions, we ran a relatively conventional mean opinion score (MOS) test, with a setup inspired by that used in the Blizzard Challenges in TTS since 2013 (see [44]). For each stimulus, we asked "How natural does the synthesised speech sound?". Responses were provided on an integer rating scale from 1 ("Completely unnatural") to 5 ("Completely natural"), with only the

[8] https://github.com/jik876/hifi-gan
[9] https://www.prolific.co/

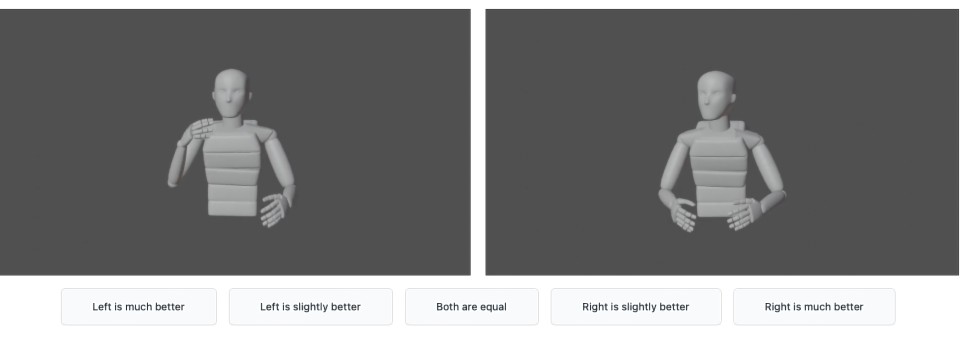

Figure 2: *Interface used in the speech-and-gesture evaluation.*

endpoints labelled. Each participant rated all stimuli from all conditions exactly once. 30 persons successfully completed the test for a total of 450 responses per system.

The results of this evaluation are shown in Table 1, column 2. All differences are statistically significant except for that between conditions G-TTS and D-TTSG. We can conclude that Diff-TTSG synthesises significantly more natural-sounding speech audio than the previous ISG state-of-the-art approach (represented by the T2-ISG baseline), and that there is no loss in quality from also learning to synthesise motion at the same time as learning to speak using Grad-TTS. At the same time, no synthetic speech was as natural as the (vocoded) original human speaker on this challenging spontaneous speech data.

### 4.5. Gesture-only evaluation

Next, we evaluated the naturalness of the gesture motion associated with the different conditions. This evaluation used video stimuli that only visualised motion, without any audio track. This ensures that participant ratings cannot be confounded by the rhythm or content of the speech, and follows the practice of recent large-scale evaluations of gesture quality (human-likeness) [29, 30]. To maintain similarity with the speech-only evaluation, participants were asked "How natural and human-like the gesture motion appears?", and gave responses on the scale 1 ("Completely unnatural") to 5 ("Completely natural"), again with only the endpoints labelled. Each participants rated all stimuli from all conditions exactly once. 30 persons successfully completed the test for a total of 450 responses per system.

The results of this evaluation are shown in Table 1, column 3. All differences are statistically significant. Diff-TTSG thus synthesises significantly more natural-looking motion than the previous state of the art, but a notable gap up to the quality of natural motion capture remains. Integrated training was better than training TTS and STG separately (D-TTSG vs. G-TTS+M). That NAT does not score higher is likely in part attributable to the limited fidelity of the motion visualisation (being upper-body only and pinned at the hip), and that the fingers, being fixed, sometimes intersect with the rest of the avatar.

### 4.6. Speech-and-gesture evaluation

For our final user study, we evaluated how appropriate the generated speech and motion were for each other. For this, we used a recent methodology from [52, 53, 30] that uses mismatching to control for the effect of overall motion quality, which otherwise can confound simple appropriateness ratings as used

Table 1: *Results of the three evaluations, showing mean opinion scores (scale: 1 to 5 or $-2$ to 2) and 95% confidence intervals.*

| Condition | Speech only | Gesture only | Speech and gesture |
|---|---|---|---|
| NAT | $4.37 \pm 0.07$ | $3.84 \pm 0.10$ | $1.20 \pm 0.10$ |
| G-TTS(+M) | $3.28 \pm 0.11$ | $2.96 \pm 0.09$ | - |
| T2-ISG | $2.91 \pm 0.12$ | $2.48 \pm 0.11$ | $0.12 \pm 0.10$ |
| D-TTSG | $3.40 \pm 0.11$ | $3.48 \pm 0.09$ | $0.44 \pm 0.10$ |

in [29, 7]. Specifically, we created a pair of video stimuli for each speech segment and condition: one video was the same as in the previous evaluation but with sound included, whereas the other stimulus had the same speech audio, but used motion from another video clip, with the motion speed slightly adjusted to make motion duration match that of the audio. This way, both videos show motion of similar quality and characteristics (since they come from the same condition), but in one video the motion is actually completely independent of the speech in the audio track. No label stated which video was which. The test then asked "Which character's motion best matches the speech, both in terms of rhythm and intonation and in terms of meaning?" The more reliably participants are able to pick out the video stimulus where the motion matches the speech, the more closely the generated motion is linked to the speech.

Based on recommendations in [31], we used five response options rather than the three (binary preference with tie) in previous work. This increases the information value of each response. It also makes passing attention checks by random chance less likely. The response options were "Left is much better", "Left is slightly better", "They are equal", "Right is slightly better", and "Right is much better".

Since each response here requires watching two videos, participants now only rated motion associated with 7 of the 15 segments (with an 8th used for the attention checks), but still rated all conditions for these segments. To get improved statistical power, we instead recruited more participants. 60 persons successfully completed the test for a total of 420 responses per system. G-TTS+M was not included in this study, since D-TTSG performed better in the gesture-only evaluation.

For analysis, the five possible responses were converted to integer values $\{-2, -1, 0, 1, 2\}$ in order, with $-2$ meaning the mismatched stimulus was rated as much better and 2 meaning the matched stimulus was rated as much better. This is similar to a CMOS test, except that the numerical values were not used to label the response options shown to participants (see

Fig. 2). A system that generates speech and audio that are specifically appropriate for each other is likely to achieve an average score significantly above zero, whereas a system that generates motion and audio that are unrelated to each other is expected to score around zero, more or less by definition.

The results of the analysis are shown in the last column of Table 1. For all conditions there is a statistical preference for matched stimuli, indicating that the two modalities are linked, both for the natural human behaviour in this data, and in the output from the two ISG systems. The proposed Diff-TTSG approach achieved significantly better differentiation of matched versus mismatched stimuli, compared to the baseline. However, consistent with the gesture-only approaches evaluated in [30], synthetic speech and motion are not as specific and appropriate for each other as their natural counterparts are. In terms of absolute numbers, a wide gap remains between synthetic and natural behaviour. Several factors may contribute to this. The models may for example be generating less distinct individual gestures than what is seen in natural gesticulation. That has been a common issue with many data-driven gesture-generation models, e.g., [54], and can act to make different stimuli in the evaluation appear more alike and indistinct. The synthesis approaches in this paper, being trained on 4.5 hours of data and having no access to word or sentence embeddings pre-trained on larger datasets, will also have difficulties in learning to make gestures that express and embody semantics, and in particular semantics consistent with the speech. Furthermore, the deterministic duration generation that Diff-TTSG inherited from Grad-TTS risks producing quite uniform speech and gesture timings, regardless of the text. Closing the gap between synthetic and natural behaviour seems like an important research target, since we may expect future improvements in the appropriateness of speech and gesture for each other, as measured by human evaluators, to correlate with more human-like multimodal behaviour and increased communicative efficacy for artificial agents.

## 5. Conclusions and future work

We have presented a new approach, based on diffusion probabilistic models, to the integrated generation of speech acoustics and 3D gesture motion from text. Compared to the previous state of the art on this multimodal synthesis task, our proposed method achieves better synthesis quality in each modality and also produces speech and gesture that are more specific and appropriate for each other. Our approach learns to speak from scratch using only 4.5 hours of spontaneous speech and gesture data. Being able to avoid the multi-stage training protocol seen in previous work was found to improve the generated motion.

Future work includes improved, stochastic duration modelling to broaden the range of different prosodic realisations that can be realised for the same text, and efforts to increase the semantic awareness of the synthesis, e.g., by modifying the approach to include text embeddings from self-supervised learning on large corpora. Other important research targets include simultaneous control over speaking and gesturing style, e.g., building on [28, 10], and incorporating additional modalities such as facial expression into the synthesis.

## 6. Acknowledgements

This work was partially supported by the Wallenberg AI, Autonomous Systems and Software Program (WASP) funded by the Knut and Alice Wallenberg Foundation, and by the Digital Futures project "Advanced Adaptive Intelligent Systems".

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
