# OpenReview forum: "Diff-TTSG: Denoising probabilistic integrated speech and gesture synthesis"
_Interspeech.org/2023/Workshop/SSW — SSW12_

### Official Review · Reviewer_uq8N · 2023-06-04
**The authors present a diffusion model for text-to-speech with synched gestures for a torso-based avatar without facial expressions. The paper is well written and presents a comprehensive evaluation of the approach. However, it could be improved by detailing and reflecting about some of the results**

**Rating:** 9
**Confidence:** 3

**Review:**

Specific Comments

Datasets and Baseline:
1. The authors claim that their approach can be trained on ‘small datasets’, but they only consider one dataset in their experiments (TSGD2 of 6h)

2. Please, include a more detailed explanation about the characteristics of the data and process followed to build the [7] baseline system, e.g., regarding the original usage of Trinity Speech-Gesture Dataset and the TSGD2

Experiments and Results:

Regarding the Speech-and-gesture evaluation:

1. “The more reliably participants pick out stimuli where the motion matches the speech, the more closely the generated motion is linked to the speech” -> How many evaluators where considered? How did you select them? Criterion?

2. “Based on recommendations in [31], we used five response options rather than the three (binary preference with tie) in previous literature. This increases the information value of each response. It also makes it harder to pass attention checks by random chance. The response options were “Left is much better”, “Left is slightly better”, “They are equal”, “Right is slightly better”, and “Right is much better”.   “For analysis, responses were converted to integer values from −2 to 2, with −2 meaning the mismatched stimulus was rated as much better and 2 meaning the matched stimulus was rated as much better, with linear interpolation in between.
-> As the authors consider MOS test, this comparative could be considered a 5-scale CMOS (Comparison MOS). Why did you decide to ‘interpolate’ the responses as they are discrete by definition?

3. Results reliability: On the one hand, the number of evaluators is different from the Speech-only and Gesture-only experiments. An on the other hand, the difference is <1 point in the 5-scale metric in all cases. Therefore, it would be interesting to reflect about the results, which, from this reviewer humble opinion, show an interesting trend, but can not be argued as a ‘strong’ conclusion of this work. For sure, this kind of subjective experiments are quite difficult to design and develop as there is not unequivocal correlation between speech and gesture, as it depends on several variables such as the speaker personality (e.g. intro vs extroversion), his/her expressiveness, etc. beyond the more clear links already observed for beat-like gestures (as shown in the demos). I would recommend including more details about this experiment and extra discussion in the paper.

Minor comments
*References:
    “CMUdict was used to phonemised text [Ref]”. -> Please, include the corresponding reference
“… 10 seconds long, which has been found to be needed for accurate evaluation of gesture synthesis [Ref]”. -> Please, include the corresponding reference
    “Example videos, to be made public on acceptance, are in the supplement and at anonymous URL bit.ly/diff-ttsg” -> This link is included twice in the paper

*Writing:
  “intends to convey [1] Gestures…” -> “intends to convey [1]. Gestures…”
  “on small datsets” -> datasets
  “are for each other Example stimuli are…” -> are for each other. Example
  “We trained a number of systems on this data” -> these data
  “Otherwise we used the same hyperparameters and stopping criterion as for G-TTS.” -> Moreover, we used… ?
  “The more reliably participants pick out..” -> “The most reliable participants…”
  “Future work includes improved, stochastic duration modelling, to broaden the range of different prosodic realisations that can be realised the same text”   Future work includes developing improved stochastic duration modelling to?

*Acronyms: SDE -> please provide its meaning (stochastic differential equation) the first time it is included in the text (section 2.1 and not in section 3.1).

---

### Official Review · Reviewer_9PF5 · 2023-06-13
**Novel and interesting paper**

**Rating:** 8
**Confidence:** 3

**Review:**

This paper proposes the first diffusion model for learning to synthesize body gestures along with speech from text.
Compared to the previous state of the art for this  multimodal synthesis task, the proposed model achieved better synthesis quality in each modality and was also able to produce more specific and mutually appropriate speech and gestures.
The proposed model is well grounded and the experiments to verify its performance are well designed.

---

### Author Response · Authors · 2023-06-28
**We have revised the final version of the paper to address reviewer comments**

We would like to express our gratitude for the detailed and constructive feedback on our submitted manuscript. For the camera-ready version of the paper, we have revised the text to clarify some of the points and to address all the points raised.

---

### Decision · Program_Chairs · 2023-06-14

**Decision:**

Accept

**Comment:**

SSW2003 received 45 papers. The acceptance rate is 82%. We are pleased to inform you that your paper has been accepted by the SSW2023 Program Committee. Please read the reviews carefully and submit your camera-ready paper by June 28th. Most reviewers performed a detailed review. Please answer to their questions and consider their comments. Note that camera-ready papers are credited with one extra page to allow authors to consider reviewers’ suggestions. So max 7 pages in total including figures & refs.
The deadline for submitting the revised version (with full non-anonymized authors and refs!) is 28th June.